# Prevalence of Stunting and Relationship between Stunting and Associated Risk Factors with Academic Achievement and Cognitive Function: A Cross-Sectional Study with South African Primary School Children

**DOI:** 10.3390/ijerph18084218

**Published:** 2021-04-16

**Authors:** Johanna Beckmann, Christin Lang, Rosa du Randt, Annelie Gresse, Kurt Z. Long, Sebastian Ludyga, Ivan Müller, Siphesihle Nqweniso, Uwe Pühse, Jürg Utzinger, Cheryl Walter, Markus Gerber

**Affiliations:** 1Department of Sport, Exercise and Health, University of Basel, 4052 Basel, Switzerland; christin.lang@unibas.ch (C.L.); sebastian.ludyga@unibas.ch (S.L.); ivan.mueller@unibas.ch (I.M.); uwe.puehse@unibas.ch (U.P.); markus.gerber@unibas.ch (M.G.); 2Department of Human Movement Science, Nelson Mandela University, Port Elizabeth 6019, South Africa; rosa.durandt@mandela.ac.za (R.d.R.); felicitas.nqweniso@mandela.ac.za (S.N.); cheryl.walter@mandela.ac.za (C.W.); 3Department of Dietetics, Nelson Mandela University, Port Elizabeth 6031, South Africa; annelie.gresse@mandela.ac.za; 4Swiss Tropical and Public Health Institute, 4051 Basel, Switzerland; kurt.long@swisstph.ch (K.Z.L.); juerg.utzinger@swisstph.ch (J.U.); 5University of Basel, 4001 Basel, Switzerland

**Keywords:** dietary diversity, food insecurity, hemoglobin, socioeconomic status, soil-transmitted helminth infections

## Abstract

Childhood stunting can have negative long-term consequences on cognitive development, academic achievement, and economic productivity later in life. We determined the prevalence of stunting and examined whether stunting and associated risk factors (low dietary diversity, insufficient hemoglobin, food insecurity, and soil-transmitted helminth (STH) infections) are associated with academic achievement and cognitive function among South African children living in marginalized communities. A cross-sectional sample of 1277 children (aged 5–12 years) was analyzed. Stunting was defined according to 2007 WHO growth references. Cognitive functioning was measured with the computerized Flanker task and academic performance via school grades. Blood and stool samples were collected to obtain hemoglobin level and STH infection. Dietary diversity was assessed by a food frequency questionnaire. Associations were examined via mixed linear regression (with school class as a random intercept). Nine percent of the children were stunted (95% CI: 7.6–10.8%). Low dietary diversity (β = 0.13, *p* = 0.004), food insecurity (β = −0.12, *p* = 0.034), and stunting (β = −0.13, *p* = 0.031) were associated with poorer end of the year results among girls. No such associations were found among boys. No significant associations were found for socioeconomic status and hemoglobin levels. The prevalence of stunting and STH infections were low in the present sample. Risk factors seem differently associated with girls’ and boys’ academic achievement. Promoting nutrition may help to promote academic achievement among girls living in low- and middle-income countries.

## 1. Introduction

For many young people in low- and middle-income countries (LMICs), it is difficult to develop their full academic potential [1]. Cognitive and academic potential is a predictor for success in real-life domains and therefore, it is important to assure the healthy development of these domains. Poor nutrition and infectious diseases can affect children’s health and educational success [2,3]. Further, cognitive function and academic performance are linked. Several studies highlighted the predictive value of executive function for both school readiness and academic achievement [4,5], which remains constant across school years [6]. Inhibitory control is conceptualized as a common component of executive function [7]. Its related abilities to resist distractions and selectively attend information directly in the classroom may partly explain the association between executive function and academic achievement. 

During early childhood, children are highly vulnerable to environmental factors like poor nutrition and diseases. Chronic malnutrition marked by stunting, has been shown to interfere with poor cognitive function in different studies [8,9]. As such, it is important to look into the extent of the lifestyle transition from a traditional to a more Westernized lifestyle, that includes changes in diet [10,11], on the South African population. The transition over the past years towards a diet with highly processed food, which is high in energy and fat, low in dietary fibre and nutritional value, is occurring among many sections of the South African population. The transitional changes, among other factors, result in a range of short- and long-term consequences, also in early life, in form of malnutrition, poor health, and wellbeing. Although some policies are already promoting early child development in the South African population and a national decrease in the prevalence of stunting has been seen [12], stunting still persist in South Africa and the prevalence is still far away from reaching the Millenium Development Goal 1: Eradicate extreme poverty and hunger. Addressing this developmental challenge will require government commitment across different departments [13]. The review by Said Mohamed et al. highlighted that there are many factors that determine stunting, including food insecurity and low socio-economic status that result in poor nutritional choices and low nutrient intake [12]. An important indicator of children’s nutritional status and health development is linear growth. Additionally, long-term consequences are described as impaired cognitive ability, economic productivity, and metabolic diseases [14]. Therefore, identifying modifiable factors that impact children’s learning capacity at school is important to establish effective interventions.

School-aged children who are stunted and those infected with soil-transmitted helminths (STHs) are at increased risk of presenting higher school absenteeism and dropout rates, and of underperforming academically [15,16] compared to well-nourished and non-infected peers. Moreover, children infected with STHs show a greater risk of being stunted [17]. However, STH infections can be regarded as an independent risk factor for children‘s cognitive function and academic achievement [18]. In contrast, several health parameters, such as hemoglobin levels, have a positive effect on cognitive function and subsequently academic achievement [19,20]. A review reported that anemic children showed lower cognitive function and academic achievement compared to their non-anemic counterparts [21]. Several studies have linked these factors, revealing a higher prevalence of STH infection and anemia among stunted children [22], and therefore making this population particularly vulnerable to poor academic achievement. 

Undernutrition is a cause of poverty because of its impact on cognitive development, and poverty is a cause of undernutrition [23]. Education is a powerful agent in disrupting this vicious cycle [24]. Moreover, undernutrition can be both a cause and a consequence of poverty, due to a lack of nutrients and economic prospects later in life [1,23]. Insufficient nutrient and energy intake have been shown to impede curiosity, interest, and learning capabilities among stunted children [19]. Despite the common assumption that the consequences of stunting are irreversible [25], recent studies highlighted physical and neurogenerative recovering from stunting. However, this process may highly depend on severity and occurrence during sensitive developmental stages in early childhood [26]. Furthermore, for reasons that are still poorly understood, stunting seems to be more prevalent among boys than girls, particularly in the lower socioeconomic strata [27].

Against this background, the aim of the present study was to estimate the prevalence of stunting, and to examine whether stunting and associated risk factors (low socioeconomic status [SES], infections with STHs, low dietary diversity, food insecurity, and low hemoglobin levels) are related to academic achievement and cognitive function among South African primary school children. Moreover, there is still a paucity of studies which have systematically addressed sex differences with respect to health outcomes and academic achievement among school-aged children living in marginalized areas from LMICs [28].

## 2. Materials and Methods 

### 2.1. Study Design

This paper is based on a cross-sectional analysis of baseline data (collected in February-April 2019) of an ongoing study in Port Elizabeth (South Africa). The *KaziAfya* study is a cluster randomized controlled trial with a 2 × 2 factorial design taking place in three African countries (Côte d’Ivoire, Tanzania, and South Africa) with a physical and multi-micronutrient intervention. Based on a-priori power analysis (for more details see [29]), we intended to recruit 1320 children in each country. 

### 2.2. Participants and Procedures

Four quintile three public schools (no-fee paying schools) were selected from peri-urban settings in Port Elizabeth. South African schools are categorized into quintiles, with quintile one representing the poorest and quintile five representing the least poor schools. Four quintile three public schools were selected from peri-urban settings in Port Elizabeth. School principals were informed about the aim, intervention, and potential risks and benefits of the study. Based on this information, the principals had the possibility to state their interest in participating in the project. 

Inclusion criteria for each school were being a public school located in a marginalized area, availability of facilities for the physical education lessons, and not registered in any other clinical trial or nutritional intervention. Children were included if they were aged below 12 years at baseline, attended grades 1–4, provided written informed consent from their parents/guardians, and did not suffer from any clinical conditions, which would prevent them from participating in the physical education lessons. A more detailed description of inclusion requirements and the study itself can be found in the study protocol [29]. The KaziAfya study was approved by the ethical review board of Northwestern and Central Switzerland (EKNZ; reference number: Req-2018–00608) and authorized by the Nelson Mandela University (NMU) ethics committee in Port Elizabeth (reference number: H18-HEA-HMS-006) and the Department of Education of the Eastern Cape Province. This study is registered in the ISRCTN registry (http://www.isrctn.com/ISRCTN29534081 (accessed on 14 April 2021). 

### 2.3. Data Assessment and Measures

The data assessment took place on the school grounds during regular school hours in February–April 2019, additionally academic achievement was collected at the end of the academic year in 2019. Data entry and validation of double entered data were performed with EpiData version 4.6.0.2 (EpiData Association; Odense, Denmark). 

### 2.4. Socioeconomic Status (SES)

Family SES was assessed by a questionnaire, which was handed out to children to take home and be answered by their caregivers. After the questionnaire was completed, the child brought it back to the classroom teacher. Nine items were used as proxy variables to assess SES, including living standards, housing characteristics, water and sanitation condition such as type of toilet and access to water, and questions addressing durable assets (i.e., electronic devices). Each of the 9 items was a dichotomized variable with 0 = not present and 1 = present. An SES score of 9 indicates a high SES. The items were summed up and averaged to build an overall SES score ranging from 0–9. In case of one missing item, a family’s mean SES score was entered into the dataset. Similar items for SES calculation were previously tested [30]. 

### 2.5. Nutritional Status 

The nutritional status of school children was determined via anthropometric data (body composition and body mass) using a digital scale (Tanita; MC-580; Tanita Corp., Tokyo, Japan) and a height measuring board to the nearest 0.1 kg and 0.1 cm, respectively. Z-scores were calculated using the WHO growth reference chart for individuals aged between 5 and 19 years [31]. A child was categorized as stunted if his/her height-for-age score was <−2 standard deviations below the median of the WHO Child Growth Standards. A standard deviation of <−3 defines severe undernutrition [32]. Categorization of stunting was defined as less than corresponding age- and sex-specific percentiles. 

### 2.6. Soil-Transmitted Helminth Infection

Stool samples were collected in small clean containers and independently examined by two clinical laboratory technicians, following the Kato-Katz procedure and then microscopically examined for STH eggs (eggs per gram of faeces = EPG) [33]. Eggs count was multiplied by 24 according to recommendations by Knopp et al. [34]. In the present study, children were classified as infected if they tested positive for either *Ascaris lumbricoides* (roundworm), *Trichuris trichiura* (whipworm), and/or *Ancylostoma duodenale* and *Necator americanus* (hookworm); not infected versus a single or multiple infection [35]. For the purpose of quality control, 10% of the stool samples were re-examined by a senior technician. If results were inconsistent with the two previous analyzed samples (κ = 0.517), a laboratory technician re-read the sample a third time. 

### 2.7. Dietary Diversity 

A dietary diversity score (DDS) was assessed by a food frequency questionnaire (FFQ) administered to the caregivers. The FFQ is based on the food composition of commonly consumed food items in South Africa [36], organized in 16 food groups; for more information see guideline for measuring dietary diversity [37]. Caregivers were asked to tick all food items which their child consumed at least once within the last 24 h. From these initial 16 food groups, an individual DDS was derived from those nine food groups that are used to calculate the Woman Dietary Diversity Score (WDDS) [37,38]: cereals and white roots and tubers; dark green leafy vegetables; vitamin A rich fruits and vegetables; other fruits and vegetables; organ meat; meat and fish; eggs; legumes, nuts and their products; and milk, milk products. Possible scores of the WDSS range between 0 and 9, with higher scores reflecting a higher dietary diversity.

### 2.8. Food Insecurity 

The Household Hunger Scale (HHS) has been specifically developed and validated for cross-cultural use [39] and to have the possibility to highlight country differences, we used the same method in three different countries (Côte d’Ivoire, Tanzania, and South Africa). Food insecurity was assessed with three questions taken from the HHS [40]. Caregivers responded to the following questions: In the past 30 days, (i) was there ever no food to eat of any kind in your house because of lack of resources to obtain food, (ii) did you or any household member go to sleep at night hungry because there was not enough food, and (iii) did you or any household member go a whole day and night without eating anything because there was not enough food? Possible answers were summed up and ranged from 0 (never) to 2 (often) [40]. Thus, the overall sum score from all three questions can vary between 0 and 6, with 0 being reflective of good access to food (household food security) and 6 being reflective of prevalence of hunger (household food insecurity). Scores between 0 and 2 are indicative that there is no or only little hunger perceived by members of a household.

### 2.9. Hemoglobin Levels

Children were pricked once into their index or middle finger. The first blood drop was swiped, the subsequent blood drops were collected into a 0.5 mL EDTA microtainer tube. For each blood test (lipid panel, HbA1c, hemoglobin (Hb), malaria, and dried blood spots) the blood was drawn out of the microtainer tube. A medical assistant collected a blood drop with a microcuvette to analyze the Hb concentration using the Hemocue Hb 301 device (HemoCue AB, Ängelholm, Sweden). The test was used to assess anemia status of each child. We defined anemia as an Hb concentration of <11.5 g/dL for children aged 5–11 years and <12.0 g/dL for children aged 12–14 years [41]. 

### 2.10. Academic Achievement

The end of year results of three subjects were used as an indicator of academic achievement: (i) home language; (ii) mathematics; and (iii) life skills. The South African school system uses a seven-point grading scale from 1 to 7, with higher scores being reflective of higher academic achievement. 

### 2.11. Cognitive Functioning 

Cognitive functioning was measured with the computerized Flanker task, which allows the assessment of information processing and inhibitory control [42]. The task was programmed and administered with E-Prime 2.0 Software (Psychology Software Tools; Sharpsburg, PA, USA). Prior to starting the test, children were seated in front of a laptop and researchers gave oral instructions. Stimuli were five white fishes presented against a black screen. The central target fish was either pointing the same direction (congruent trial) or different direction (incongruent trial) than the flanking fish. Children were asked to focus on the central fish and indicate its direction by pressing a button corresponding to left or right. Accuracy and response speed were equally emphasized. After two practice rounds with 60 trials in total, two blocks (interspersed by 30 s) with 40 trials each were administered. Congruent and incongruent trials were presented with equal probability and in a randomized order. After onset of the visual stimuli, participants had 2500 ms to respond. The inter-trial interval varied between 1100 and 1500 ms to reduce the likeliness of guessing. Performance indices calculated from the Flanker task were mean accuracy and (response-correct) reaction time for congruent and incongruent trials to assess information processing and inhibitory control, respectively. To ensure that only participants understanding the task were included in analyses, datasets with accuracy rates lower than chance (≤50%) on the second practice round, were removed. 

### 2.12. Statistical Analyses 

Descriptive statistics were calculated with SPSS version 26 for Mac (IBM Corporation; Armonk, NY, USA). SPSS was also used to test sex differences via univariate analyses of variance (ANOVAs) and χ^2^-tests, as well as correlations between academic achievement and cognitive function. After detection of considerable sex differences (see Results section for more details) with regard to academic achievement and cognitive function, it was decided to perform separate analyses for boys and girls. To examine whether and to what degree stunting and associated risk factors (low SES, infections with STH, low dietary diversity, food insecurity, and low Hb levels) are related to academic achievement and cognitive function (after controlling for age and zBMI), a series of mixed linear regression analyses were calculated, using Mplus software version 7, (Muthén & Muthén; Los Angeles, CA, USA, 1998–2020). Mixed linear regression models were chosen because they can take into account the nested nature of the data. This seemed important in the present study because academic achievement can strongly vary as a function of school and class. In our paper, we therefore included school class as a random intercept across all analyses. Model estimation was based on a robust maximum likelihood estimator (MLR). Since missing data were handled via full information maximum likelihood (FIML), all mixed linear regression analyses are based on the full sample. Little’s Missing Completely at Random (MCAR) test was applied to examine whether all prerequisites of FIML were met. Regarding cognitive performance on the Flanker task, the analyses were repeated after exclusion of children who did not reach sufficient accuracy levels (≤50%) in the second practice trial of the Flanker task to rule out possible by chance responses. The estimate (standardized Beta-weight), standard error (S.E.) of the estimate, and *p*-value are displayed to describe the results of the mixed linear regression analyses. For all statistical analyses, the level of significance was set at *p* < 0.05.

## 3. Results

### 3.1. Sample Characteristics and Inspection of Missing Data

A total of 1369 children provided written informed consent. Hereof, 1277 took part in the baseline data assessment and presented with valid data regarding their sex. The final sample consisted of 613 girls (48.0%) and 664 boys (52.0%). Considerable variation was observed across variables regarding the number of missing data (Table 1). Little’s MCAR test indicated that data were missing completely at random, and that therefore the prerequisites to use FIML were fulfilled, χ^2^ (df = 757) = 709.4, *p* = 0.891.

### 3.2. Descriptive Data and Prevalence of Stunting and STH Infections

As shown in Table 1, the mean age of children was 8.3 (±1.4) years and the mean BMI was 16.1 (±2.6) m/kg^2^. In total, 7.2% of the study population were infected with STH (95% CI: 5.7–8.5) and 9.1% of the children were categorized as stunted (95% CI: 7.6–10.8). Higher proportions of children were reported to be anemic (19.2%, 95% CI: 16.7–21.7) with a mean hemoglobin concentration of 12.3 (±0.9) g/dl. The analyses found that the mean food insecurity score was 1.8 (±1.2) and mean dietary diversity score reached 5.8 (±2.0). The mean of end of the year result was 4.8 (±1.3), whereas the mean accuracy levels in the Flanker task was 92.8% (±10.8) for congruent stimuli and 84.9% (±19.4) for incongruent stimuli.

### 3.3. Sex Differences

Table 1 shows that boys and girls did not differ with regard to their height, weight, BMI, SES, dietary diversity, and food insecurity. Significant sex differences were found for age (*p* < 0.05), with girls being slightly younger (8.2 ± 1.4 years) than boys (8.4 ± 1.5 years). Although more girls (10.7%) were stunted than boys (7.6%), the difference was not statistically significant (*p* = 0.057). No significant sex differences were found with regard to STH infections (*p* > 0.05). Stratification by sex revealed significant differences with regard to overall academic achievement (end of year results, *p* < 0.001) and across all subjects (language, mathematics, and life skills; all *p* < 0.001), with girls achieving higher grades than boys. In the Flanker task, accuracy levels were similar in boys and girls. Nevertheless, faster reaction times were observed among boys, both for congruent and incongruent stimuli (*p* < 0.001). 

### 3.4. Correlations between Academic Achievement and Cognitive Function

As shown in Table 2, several significant, but weak (*r* ≤ 0.30) correlations were found between academic achievement and the Flanker task performance indicators. With regard to accuracy, the bivariate correlations between academic achievement and the Flanker task performance indicators were slightly stronger in girls (*r* = 0.19 to 0.30) than in boys (*r* = 0.12 to 0.19). Similarly, stronger correlations were found between academic achievement and reaction times to the Flanker task stimuli in girls (*r* = −0.09 to −0.22) than in boys (*r* = −0.02 to −0.11). Among boys, the correlations with academic achievement were not significant if incongruent Flanker task stimuli were used. Correlations between different school subjects varied between *r* = 0.77 and 0.88 (*p* < 0.001) in girls, and between *r* = 0.72 and 0.83 in boys (*p* < 0.001). Only weak correlations were observed between accuracy and reaction times in the Flanker task, both for congruent (girls: *r* = −0.17, *p* < 0.001; boys: *r* = −0.11, *p* < 0.05) and incongruent stimuli (girls: *r* = −0.17, *p* < 0.001; boys: *r* = 0.13, *p* < 0.001).

### 3.5. Associations with Academic Achievement

The mixed multiple linear regression analyses showed that among boys, neither stunting nor any other of the assessed factors were associated with academic achievement (Table 3). Among girls, the analyses showed that stunted girls achieved significantly lower end of the year results (*p* < 0.05) and showed significantly lower performances in mathematics (*p* < 0.05). Although the associations pointed in the same direction for language and life skills, the relationships were not statistically significant. Among girls, two further significant relationships were observed. First, higher food diversity was associated with better academic achievements across all indicators (*p* < 0.05). Second, food insecurity was associated with lower end of the year results and lower grades in language (*p* < 0.05). Although the relationships pointed in the same directions for mathematics and life skills, the associations were not statistically significant.

### 3.6. Associations with Cognitive Function

Table 4 shows that among boys, age was associated with better congruent and incongruent stimuli (*p* < 0.001). Thus, older children achieved higher accuracy scores and faster reaction times in the Flanker task. After excluding boys from the analyses who did not perform better than chance (≤50% accuracy), accuracy was no longer significantly associated with the reaction times in the Flanker task. When incongruent stimuli were presented, stunted boys had slightly higher reaction times in the Flanker task than their non-stunted counterparts.

Among girls, age was also associated with better congruent and incongruent stimuli (*p* < 0.001) (see Table 5). Again, older girls provided more accurate and faster responses to congruent and incongruent stimuli. As for boys, girls’ accuracy and reaction times were not associated with each other. Being infected with STH was associated with less accurate responses to congruent Flanker task stimuli (*p* < 0.05), whereas being stunted was related to higher reaction times in response to congruent Flanker task stimuli.

## 4. Discussion

The key findings of the present study are that low DDS, food insecurity and being stunted are negatively associated with girls’ end of the year results. Moreover, in girls, being infected with STH and being stunted were related with slower reaction times in response to congruent stimuli in the Flanker task. By contrast, no such associations were found for boys. Irrespective of children‘s sex, neither SES nor Hb, were significantly associated with their academic performance and cognitive function. Thus, the findings from this study suggest that risk factors might be differently associated with girls’ and boys’ academic achievement and cognitive function. 

In the present sample, 9.1% of the children were stunted, with more girls being stunted than boys. The result is consistent with findings of a previous study [43], in which a similar prevalence of 12.3% was observed among South African children living in marginalized areas. As highlighted by Casale and Desmond [26], catch up in height is more likely among younger than older children. As stunting may not persist throughout life, we can assume that the number of stunted children will decrease as they grow older. 

We found that 7.1% of the children were infected with STHs. This prevalence is lower than the one observed by Gall et al. (2017), where 31.1% of the children were infected with STH before a nationally initiated deworming program was implemented [43,44]. Nevertheless, a previous study also showed that even within the same geographical area, there might be considerable variation between schools located in different neighborhoods [45]. For instance, the authors [46] showed in a study with eight schools in the Port Elizabeth region (South Africa) that STH infection rates climbed up to 70% in two schools, whereas they were lower in the other schools. However, Müller et al. [46] also observed that the STH infection rate can be successfully reduced via regular administration of deworming medication and complementing water, sanitation, and hygiene (WASH) interventions, hereby highlighting the potential impact of a national deworming program on children’s growth and wellbeing.

With regard to nutrition, our study showed that the mean DDS was 5.8 (±2.0) out of nine possible food groups (with a maximum score of 9). Poor dietary diversity is defined as WDDS being lower than 4, which was not the case for most children in our sample (93.7%). Moreover, using the same instrument as we did in our study, Labadarios et al. [47] found that at the national level, the mean dietary diversity score was 4.0 among 16- year-old South Africans. Again, this suggests that the diet was relatively different in our population, despite the fact that all children were living in marginalized areas. Labadarios et al. further suggested that dietary diversity can be seen as a proxy indicator for food insecurity. In line with this, mean food insecurity was relatively low (M = 1.8 ± 1.2) in our study, which accords well with a previous study carried out in the same geographic area [43]. Nevertheless, additional analyses showed that the correlations between dietary diversity and food insecurity were relatively low in the present sample (girls and boys: *r* = 0.12, *p* < 0.05), which suggests that in our sample dietary diversity and food insecurity assess different aspects of nutritional behavior. The fact that we found relatively high dietary diversity and low food insecurity is a positive finding. First, some authors claim that although there is a decrease in food insecurity in South Africa, food insecurity is still a public health challenge [48,49]. The Statistics South Africa highlighted that just 41.4% of 5– to 12– year-old children are food secure in South Africa. Second, having access to food can have an important impact on how well a child is nourished. In line with this notion, the National Food Consumption Survey (NFCS) of South African children aged 1-9 years showed a strong correlation between food consumption and growth impairment [48]. 

Furthermore, in this study, significant differences were observed between boys and girls in academic achievement. These disparities might be attributable to a sex-specific rhythm of cognitive, physical and psychomotor development [50,51]. In our sample, girls generally achieved better school grades than their male peers. Many possible reasons and mechanisms have been proposed to explain these sex differences. For instance, studies revealed that girls are more likely to be accepted by teachers and so have a greater advantage to express themselves verbally during primary school-age, while boys are missing male role models in school [52,53]. Roivainen [48] and Siedlecki et al. [49] further suggested that the development of verbal skills is faster among girls during early childhood as compared to their male counterparts. By contrast, our study revealed that if compared to girls, boys achieved similar accuracy rates, but faster reaction times in the Flanker task. Thus, male superiority in reaction time and attention tests seem to be based on different factors than female superiority in academic achievement. 

With regard to the associations between stunting, academic achievement and cognitive function, our results revealed that stunting is negatively associated with academic achievement and cognitive function. This accords well with previous research [54]. One reason may be that stunted children are more absent from school, which causes higher disruption in academic learning [55]. Moreover, stunted children suffer chronically from low energy levels and nutrient intake, which can lead to a reduced interest for learning [19]. Few studies, however, have examined the relationship between stunting and academic achievement among South African school children. For instance, Pienaar found that height-for-age score was significantly correlated with academic performances and motor functioning skills [56]. A unique finding of our study was that stunting was only associated with lower end of the year results among girls and that stunting was not significantly associated with all school subjects. In line with our study, a significant correlation was observed between stunting and main school subjects in Malaysian school-aged children [57]. The Malaysian study similarly found that girls achieved better academic performances than boys. A more recent study found that academic achievement is particularly sensitive to changes in nutritional availability among stunted girls [58]. Overall, it is known that an early onset of nutritional deficits can have persistent effects on children’s health, academic achievement, and cognitive performance [9,54,59,60]. Micronutrient deficiencies can result in short- and long-term alterations of brain development [61]. Vitamin A, iron, zinc, and iodine deficiencies are the most investigated nutrients, showing that a lack of those contributes to growth and intellectual impairments [62]. In line with this, we found that stunting was significantly associated with lower reaction time in the Flanker task.

Sufficient nutrient availability is a basic requirement for optimal brain development and cognitive function over time [63]. Due to a lifestyle and nutrition transition to a westernized lifestyle and diet, including calorie-rich but nutrient poor dense food, LMICs underlie a double burden. This includes both childhood undernutrition for households which still have low access to a diverse diet and the rising prevalence of overweight due to highly processed food consumption and growing physical inactivity. Meo et al. observed a significantly reduced attention span, intelligence, and cognitive flexibility among severely obese children if compared to children of normal weight [64]. However, Afzal and Gortmaker reported that there is conflicting evidence regarding a direct link between childhood obesity and cognitive function and that a lower cognitive function in obese children is often due to other accompanying psychosocial and physiological factors [65]. As emphasized by Howard [54], food security is important from a public health perspective. Thus, even if food insecurity decreases throughout primary school years, children still suffer from the consequences up to many years later. Raskind et al. have argued that the effects of food insecurity on academic achievement might be particularly strong among girls because they seem to be more sensitive to changes in their family environment [66]. The systematic literature review by Jung et al. supports that women are particularly vulnerable to food insecurity [58]. In line with this notion, our analyses suggest that food security and dietary diversity are more closely associated with higher academic achievement among girls. This sheds new light on possible sex-related differences regarding the interplay between nutrition, health, and wellbeing among school-aged children. 

Our study has several strengths. First, we considered stunting and various related risk factors in our mixed regression models, which might be independently associated with academic achievement and cognitive function and which could be potential targets for future interventions. Second, findings are based on a relatively large sample (*N* = 1277) of South African children stemming from marginalized communities. In these settings, the population is particularly vulnerable to factors like stunting, food insecurity, and low dietary diversity, combined with other health-related risk factors, and adequate academic achievement is particularly important for children to break the vicious circle of poverty and poor health. Third, we used mixed linear regression analyses with an intercept for school-classes to take into account the nested (clustered) nature of our data. Fourth, performing separate analyses for boys and girls allowed us to gain new insights into potential factors underlying sex differences in academic achievement and cognitive function. Fifth, we also examined bivariate correlations between explanatory variables. As the strongest correlation was relatively weak (*r* = −0.27, *p* < 0.001; between SES and food insecurity) and as all variance inflation factors were low (VIF ≤ 1.2), we feel confident that multi-collinearity was not an issue in the present study. Despite these strengths, some limitations should be noted. First, due to the cross-sectional nature of the data, conclusions about cause and effect are not possible. Second, a special focus was placed on food variety, whereas we did not assess information about quantity of food items. Third, children had different experiences with laptop applications, which might have had an impact on the Flanker task performances. Fourth, findings from the Flanker task cannot be generalized to overall cognitive ability, because the test covers only two specific aspects (attention and inhibitory control). Fifth, we obtained only one stool sample, hence underestimating STH infection prevalence. Sixth, we also acknowledge that our sample is not representative for South African children across all the SES. This is important as access to varied food might be less limited among children attending wealthier schools, or might be even more restricted among learners living in rural settings. Seventh, the cross-sectional nature of our data did not allow us to further examine the pathways by which the risk factors considered in this analysis may interact. The investigation of such mediator effects is strongly recommended for future (longitudinal) research. Eighth, extensive government and NGO efforts have been devoted to fighting childhood malnutrition in South Africa. Therefore, it would be interesting to compare student outcomes from local schools with different food environment. For example, do free/subsidized meal programs help relieve food insecurity, increase dietary diversity and improve the performance of children from low- and middle-income families. Nevertheless, while the results of our on-going randomized controlled trial will show whether cognitive performance can be improved via multi-micronutrient supplementation, a different research design would be needed to assess the impact of governmental and NGO measures.

## 5. Conclusions

While the prevalence of stunting and STH infections was relatively low in the present sample, we found that stunting and related risk factors might be differently associated with girls’ and boys’ academic achievement. We encourage researchers to replicate our study and to focus more closely on nutritional aspects and sex differences in the study of academic achievement and cognitive function. Our findings suggest that low dietary diversity, food insecurity, and being stunted may have particularly negative effects on academic achievement and cognitive function among girls. Accordingly, improving nutrition may be particularly suitable to promote academic achievement among girls. On the other hand, our study also suggest that other factors need to be identified in order to improve academic attainment and cognitive function among boys.

## Figures and Tables

**Table 1 ijerph-18-04218-t001:** Baseline characteristics of the South African study population in February-April 2019.

Measures	*N*	All Children (*n* = 1277)	Boys (*n* = 664)	Girls (*n* = 613)	Sex Differences (Girls = 0, Boys = 1)		
						*p*-Value	η^2^
Age and anthropometry		***M* (*SD*)**	***M* (*SD*)**	***M* (*SD*)**	***F***		
Age (years)	1271	8.3 (1.4)	8.4 (1.5)	8.2 (1.4)	6.2	0.013	0.005
Height (cm)	1240	124.7 (9.2)	125.1 (9.0)	124.3 (9.4)	2.9	0.087	0.002
Weight (kg)	1240	25.4 (6.9)	25.4 (6.4)	25.4 (7.2)	0.0	0.884	0.000
BMI (kg/m^2^)	1240	16.1 (2.6)	16.0 (2.6)	16.1 (2.8)	0.5	0.470	0.000
zBMI	1240	−0.1 (1.2)	−0.1 (1.0)	−0.1 (1.4)	0.0	0.843	0.000
		***n* (%)**	***n* (%)**	***n* (%)**	**χ^2^**		
Stunted	1231	112 (9.1)	48 (7.6)	64 (10.7)	3.6	0.057	
Sociocultural characteristics		***M* (*SD*)**	***M* (*SD*)**	***M* (*SD*)**	***F***		
Socioeconomic status (SES)	855	6.8 (1.4)	6.8 (1.4)	6.8 (1.4)	0.0	0.825	0.000
Soil-transmitted helminth (STH) infections		***n* (%)**	***n* (%)**	***n* (%)**	**χ^2^**		
Infected	1245	89 (7.1)	50 (7.8)	39 (6.5)	0.8	0.383	
Nutrition		***M* (*SD*)**	***M* (*SD*)**	***M* (*SD*)**	***F***		
Dietary diversity (WDDS)	943	5.8 (2.0)	5.8 (2.0)	5.8 (2.0)	0.1	0.812	0.000
Food insecurity (hunger scale)	894	1.8 (1.2)	1.8 (1.2)	1.8 (1.2)	0.0	0.985	0.000
Hemoglobin concentration and anemia		***M* (*SD*)**	***M* (*SD*)**	***M* (*SD*)**	***F***		
Hemoglobin level (g/dL)	981	12.3 (0.9)	12.2 (0.9)	12.4 (0.9)	8.9	0.003	0.009
		***n* (%)**	***n* (%)**	***n* (%)**	**χ^2^**		
Anaemic	975	187 (19.2)	105 (21.6)	82 (16.8)	3.7	0.055	
Academic achievement		***M* (*SD*)**	***M* (*SD*)**	***M* (*SD*)**	***F***		
End of year results	1045	4.6 (1.3)	4.4 (1.2)	4.8 (1.3)	35.9	0.000	0.033
Language	1045	4.5 (1.3)	4.3 (1.2)	4.8 (1.3)	36.3	0.000	0.034
Mathematics	1045	4.7 (1.3)	4.4 (1.2)	4.9 (1.4)	30.4	0.000	0.028
Life skills	1045	5.0 (1.0)	4.8 (1.0)	5.2 (1.0)	39.7	0.000	0.037
Flanker task		***M* (*SD*)**	***M* (*SD*)**	***M* (*SD*)**	***F***		
Accuracy (congruent stimuli)	1223	92.8 (10.8)	92.6 (11.1)	92.9 (10.4)	0.2	0.651	0.000
Accuracy (incongruent stimuli)	1223	84.9 (19.4)	85.4 (18.2)	84.4 (20.6)	0.07	0.397	0.001
Reaction time (congruent stimuli)	1223	1187.9 (240.4)	1155.7 (243.5)	1222.2 (232.3)	23.9	0.000	0.019
Reaction time (incongruent stimuli)	1223	1273.6 (266.6)	1238.6 (262.5)	1310.9 (266.0)	22.7	0.000	0.018

Notes. STH infections = Single or multiple infection with soil-transmitted helminths.

**Table 2 ijerph-18-04218-t002:** Correlations between indicators of academic achievement (school grades) and cognitive function (Flanker task), among South African boys (*n* = 664) and girls (*n* = 613) in February- April 2019.

Measures	Accuracy (Congruent Stimuli)	Accuracy (Incongruent Stimuli)	Reaction Time (Congruent Stimuli) ^a^	Reaction Time (Incongruent Stimuli) ^b^
	*r*	*r*	*r*	*r*
Boys				
End of year results	0.15 **	0.17 **	−0.10 *	−0.04
Language	0.12 *	0.14 *	−0.08	−0.02
Mathematics	0.18 ***	0.19 ***	−0.11 *	−0.05
Life skills	0.17 ***	0.19 ***	−0.11 *	−0.05
Girls				
End of year results	0.21 ***	0.30 ***	−0.17 ***	−0.10 *
Language	0.19 ***	0.28 ***	−0.18 ***	−0.12 *
Mathematics	0.22 ***	0.30 ***	−0.15 **	−0.09 *
Life skills	0.23 ***	0.30 ***	−0.22 ***	−0.16 **

Notes. ^a^ Controlled for accuracy (congruent stimuli). ^b^ Controlled for accuracy (incongruent stimuli). * *p* < 0.05. ** *p* < 0.01. *** *p* < 0.001.

**Table 3 ijerph-18-04218-t003:** Mixed multiple linear regression analyses to explain academic achievement, separately for South African boys (*n* = 664) and girls (*n* = 613).

Explanatory Variables	Mixed Multiple Linear Regression					
	Boys			Girls		
	Estimate	*S.E.*	*p*-Value	Estimate	*S.E.*	*p*-Value
End of the year results						
Age (years)	−0.01	0.05	0.892	−0.10	0.07	0.137
zBMI	0.03	0.05	0.524	0.03	0.05	0.508
Stunting (0 = not stunted, 1 = stunted) *	−0.04	0.05	0.382	**−0.13**	**0.06**	**0.031**
Socioeconomic status (SES)	−0.10	0.06	0.138	−0.06	0.05	0.299
Hemoglobin level (g/dL)	0.01	0.05	0.847	0.04	0.05	0.442
STH infection (0 = not infected, 1 = infected) *	0.02	0.04	0.514	−0.09	0.05	0.072
Food insecurity (hunger scale)	0.04	0.06	0.573	**−0.12**	**0.05**	**0.034**
Dietary diversity (WDDS)	0.00	0.06	0.978	**0.13**	**0.04**	**0.004**
Language						
Age (years)	−0.06	0.06	0.267	−0.13	0.07	0.072
zBMI	0.03	0.05	0.570	0.06	0.05	0.218
Stunting (0 = not stunted, 1 = stunted) *	−0.05	0.04	0.296	−0.10	0.06	0.073
Socioeconomic status (SES)	0.08	0.07	0.240	−0.07	0.05	0.202
Hemoglobin level (g/dL)	0.02	0.05	0.661	0.04	0.05	0.427
STH infection (0 = not infected, 1 = infected) *	−0.02	0.04	0.707	−0.09	0.05	0.101
Food insecurity (hunger scale)	0.04	0.06	0.484	**−0.12**	**0.06**	**0.026**
Dietary diversity (WDDS)	0.03	0.06	0.665	**0.14**	**0.04**	**0.002**
Mathematics						
Age (years)	0.05	0.05	0.361	−0.07	0.06	0.282
zBMI	0.03	0.05	0.499	0.00	0.05	0.946
Stunting (0 = not stunted, 1 = stunted) *	−0.03	0.05	0.503	**−0.14**	**0.06**	**0.015**
Socioeconomic status (SES)	0.10	0.06	0.096	−0.04	0.06	0.441
Hemoglobin level (g/dL)	0.00	0.06	0.956	0.03	0.04	0.493
STH infection (0 = not infected, 1 = infected) *	0.06	0.04	0.077	−0.09	0.05	0.071
Food insecurity (hunger scale)	0.03	0.07	0.691	−0.10	0.05	0.061
Dietary diversity (WDDS)	−0.03	0.06	0.595	**0.11**	**0.05**	**0.016**
Life skills						
Age (years)	0.06	0.06	0.369	0.08	0.08	0.337
zBMI	0.03	0.05	0.572	0.00	0.05	0.938
Stunting (0 = not stunted, 1 = stunted) *	−0.10	0.05	0.057	−0.10	0.06	0.119
Socioeconomic status (SES)	0.17	0.05	0.002	−0.05	0.06	0.373
Hemoglobin level (g/dL)	−0.01	0.05	0.849	0.04	0.04	0.399
STH infection (0 = not infected, 1 = infected) *	−0.01	0.03	0.652	−0.09	0.05	0.090
Food insecurity (hunger scale)	0.08	0.06	0.168	−0.09	0.05	0.059
Dietary diversity (WDDS)	0.01	0.06	0.841	**−0.10**	**0.05**	**0.035**

Notes. All analyses controlled for class-in-school (random intercept). zBMI = *z*-standardized body mass index scores. SES = Socioeconomic status. STH = Soil-transmitted helminth. STH infection = Single or multiple infection. WDDS = Women Dietary Diversity Score. * Being “not stunted” and being “not infected” are used as reference. Statistically significant associations (*p* < 0.05) are displayed in bold format.

**Table 4 ijerph-18-04218-t004:** Multiple mixed linear regression analyses to Flanker task results of South African boys (*n* = 664).

Explanatory Variables	Mixed Multiple Linear Regression					
	All Boys (*n* = 664)			After Exclusion of Boys Who Did Not Perform Higher than Chance (*n* = 641)		
	Estimate	*S.E.*	*p*-Value	Estimate	*S.E.*	*p*-Value
Flanker Task: Accuracy (Congruent Stimuli)						
Age (years)	**0.31**	**0.04**	**0.000**	**0.30**	**0.04**	**0.000**
zBMI	−0.06	0.04	0.166	−0.07	0.04	0.104
Stunting (0 = not stunted, 1 = stunted) *	−0.03	0.04	0.411	−0.04	0.04	0.352
Socioeconomic status (SES)	0.00	0.03	0.988	0.00	0.03	0.943
Hemoglobin level (g/dL)	0.01	0.05	0.795	0.01	0.06	0.859
STH infection (0 = not infected, 1 = infected) *	0.01	0.02	0.511	0.01	0.02	0.693
Food insecurity (hunger scale)	−0.05	0.04	0.255	−0.05	0.04	0.252
Dietary diversity (WDDS)	0.08	0.05	0.100	0.09	0.05	0.094
Flanker task: Accuracy (incongruent stimuli)						
Age (years)	**0.25**	**0.07**	**0.000**	**0.24**	**0.07**	**0.000**
zBMI	−0.04	0.04	0.319	−0.05	0.04	0.192
Stunting (0 = not stunted, 1 = stunted) *	−0.01	0.04	0.793	−0.01	0.04	0.742
Socioeconomic status (SES)	0.04	0.05	0.377	0.04	0.05	0.389
Hemoglobin level (g/dL)	0.02	0.04	0.670	0.02	0.04	0.649
STH infection (0 = not infected, 1 = infected) *	0.05	0.03	0.091	0.05	0.03	0.110
Food insecurity (hunger scale)	0.00	0.06	0.946	0.01	0.06	0.942
Dietary diversity (WDDS)	−0.01	0.04	0.884	0.00	0.04	0.972
Flanker task: Reaction time (congruent stimuli)						
Accuracy in congruent stimuli	**−0.08**	**0.04**	**0.023**	0.05	0.04	0.219
Age (years)	**−0.51**	**0.04**	**0.000**	**−0.55**	**0.04**	**0.000**
zBMI	−0.01	0.03	0.712	−0.01	0.03	0.814
Stunting (0 = not stunted, 1 = stunted) *	0.04	0.03	0.175	0.05	0.03	0.164
Socioeconomic status (SES)	−0.05	0.04	0.225	−0.06	0.04	0.163
Hemoglobin level (g/dL)	0.01	0.04	0.828	0.01	0.04	0.804
STH infection (0 = not infected, 1 = infected) *	0.01	0.03	0.876	0.00	0.03	0.997
Food insecurity (hunger scale)	0.04	0.05	0.401	0.04	0.05	0.411
Dietary diversity (WDDS)	−0.02	0.05	0.680	−0.02	0.05	0.655
Flanker task: Reaction time (incongruent stimuli)						
Accuracy in incongruent stimuli	−0.02	0.04	0.634	−0.02	0.04	0.592
Age (years)	**−0.42**	**0.04**	**0.000**	**−0.42**	**0.04**	**0.000**
zBMI	0.00	0.04	0.957	0.00	0.04	0.984
Stunting (0 = not stunted, 1 = stunted) *	**0.08**	**0.03**	**0.021**	**0.08**	**0.03**	**0.025**
Socioeconomic status (SES)	−0.04	0.05	0.470	−0.03	0.05	0.495
Hemoglobin level (g/dL)	0.00	0.04	0.968	0.00	0.04	0.997
STH infection (0 = not infected, 1 = infected) *	0.00	0.03	0.903	0.00	0.04	0.947
Food insecurity (hunger scale)	0.01	0.06	0.922	0.01	0.06	0.920
Dietary diversity (WDDS)	0.05	0.05	0.289	0.05	0.05	0.288

Notes. All analyses controlled for class-in-school (random intercept). zBMI = *z*-standardized body mass index scores. SES = Socioeconomic status. STH = Soil-transmitted helminth. STH infection = Single or multiple infection. WDDS = Women Dietary Diversity Score. * Being “not stunted” and being “not infected” are used as reference. Statistically significant associations (*p* < 0.05) are displayed in bold format.

**Table 5 ijerph-18-04218-t005:** Multiple mixed linear regression analyses to explain Flanker task results of South African girls (*n* = 613).

Explanatory Variables	Mixed Multiple Linear Regression					
	All Girls (*n* = 613)			After Exclusion of Girls Who Did Not Perform Higher than Chance (*n* = 598)		
	Estimate	*S.E.*	*p*-Value	Estimate	*S.E.*	*p*-Value
Flanker task: Accuracy (congruent stimuli)						
Age (years)	**0.22**	**0.05**	**0.000**	**0.22**	**0.05**	**0.000**
zBMI	0.00	0.05	0.935	−0.01	0.05	0.828
Stunting (0 = not stunted, 1 = stunted) *	0.01	0.05	0.870	0.00	0.05	0.954
Socioeconomic status (SES)	0.09	0.07	0.175	0.08	0.07	0.250
Hemoglobin level (g/dL)	0.07	0.04	0.125	0.07	0.04	0.113
STH infection (0 = not infected, 1 = infected) *	**−0.16**	**0.06**	**0.015**	**−0.15**	**0.06**	**0.016**
Food insecurity (hunger scale)	−0.02	0.05	0.748	−0.03	0.05	0.619
Dietary diversity (WDDS)	−0.01	0.03	0.829	0.00	0.03	0.951
Flanker task: Accuracy (incongruent stimuli)						
Age (years)	**0.23**	**0.06**	**0.000**	**0.23**	**0.06**	**0.000**
zBMI	−0.09	0.05	0.066	−0.09	0.05	0.054
Stunting (0 = not stunted, 1 = stunted) *	−0.11	0.07	0.107	−0.11	0.07	0.096
Socioeconomic status (SES)	0.00	0.04	0.966	−0.01	0.04	0.896
Hemoglobin level (g/dL)	0.00	0.04	0.097	0.00	0.04	0.982
STH infection (0 = not infected, 1 = infected) *	−0.10	0.06	0.089	−0.10	0.06	0.084
Food insecurity (hunger scale)	0.04	0.04	0.349	0.03	0.04	0.432
Dietary diversity (WDDS)	−0.02	0.04	0.659	−0.02	0.04	0.715
Flanker task: Reaction time (congruent stimuli)						
Accuracy in congruent stimuli	**−0.10**	**0.03**	**0.000**	−0.06	0.03	0.095
Age (years)	**−0.44**	**0.04**	**0.000**	**−0.46**	**0.04**	**0.000**
zBMI	−0.05	0.03	0.126	−0.04	0.03	0.189
Stunting (0 = not stunted, 1 = stunted) *	**0.09**	**0.03**	**0.006**	**0.10**	**0.04**	**0.003**
Socioeconomic status (SES)	0.06	0.05	0.194	0.07	0.05	0.160
Hemoglobin level (g/dL)	−0.03	0.04	0.420	−0.03	0.04	0.460
STH infection (0 = not infected, 1 = infected) *	0.02	0.03	0.542	0.02	0.03	0.521
Food insecurity (hunger scale)	0.00	0.05	0.995	0.00	0.05	0.961
Dietary diversity (WDDS)	−0.03	0.04	0.319	−0.03	0.04	0.347
Flanker task: Reaction time (incongruent stimuli)						
Accuracy in incongruent stimuli	−0.07	0.05	0.154	−0.08	0.05	0.130
Age (years)	**−0.36**	**0.04**	**0.000**	**−0.36**	**0.04**	**0.000**
zBMI	−0.02	0.04	0.667	−0.02	0.04	0.607
Stunting (0 = not stunted, 1 = stunted) *	0.04	0.04	0.294	0.04	0.04	0.323
Socioeconomic status (SES)	0.02	0.05	0.750	0.02	0.05	0.754
Hemoglobin level (g/dL)	−0.04	0.04	0.413	−0.04	0.04	0.396
STH infection (0 = not infected, 1 = infected) *	0.03	0.04	0.468	0.03	0.04	0.484
Food insecurity (hunger scale)	−0.03	0.05	0.617	−0.02	0.05	0.617
Dietary diversity (WDDS)	0.00	0.04	0.937	0.01	0.04	0.915

Notes. School class was used as a random intercept across all analyses. zBMI = *z*-standardized body mass index scores. SES = Socioeconomic status. STH = Soil-transmitted helminth. STH infection = Single or multiple infection. WDDS = Women Dietary Diversity Score. * Being “not stunted” and being “not infected” are used as reference. Statistically significant associations (*p* < 0.05) are displayed in bold format.

## Data Availability

All analyzed data generated during this study are available on reasonable request.

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
