# Peer review of "Prevalence of Stunting and Relationship between Stunting and Associated Risk Factors with Academic Achievement and Cognitive Function: A Cross-Sectional Study with South African Primary School Children"

_ijerph, 2021, doi:10.3390/ijerph18084218_

Round 1
Reviewer 1 Report
Authors have addressed all comments.
Author Response
Thank you again for your appreciative feedback.
Reviewer 2 Report
MAJOR COMMENTS:
The authors have now addressed potential multicollinearity analytically (by reporting correlation coefficients and VIF statistics). This is sufficient. It is also understandable that they do not wish to pursue a formal mediation manuscript in this manuscript. Nevertheless, it is a missed opportunity for the authors to strengthen the discussion by at least discussing theoretical linkages between coefficients, to better contextualize their motivations in performing the analysis, and their results.
The choice of test and measurement does not need to be explained to me, the reviewer. Rather it should be clarified in the manuscript. Why did you choose to assess information processing and inhibitory control as opposed to other domains that might be measured such as cognitive flexibility, working memory, or others? Presumably, they are related to school performance or other key outcomes? Or perhaps these are key domains that develop during this age? You added a bit of this to the introduction (lines 42-43), further background in the methods would also be helpful. Keep in mind that the readership of this journal includes environmental scientists, tropical medicine physicians, and others, who may not be familiar with the field of cognition, orienting these readers is helpful.
MINOR COMMENTS:
TABLE 1- Why is ‘stunting’ listed under ‘STHs, stunting, and anemia” rather than ‘Age & Anthropometry” (Stunting is an anthropometry measure?) Why is hemoglobin listed apart from anemia, shouldn’t they be grouped?
LINE 43-44- It seems that this should be 2 paragraphs, one about cognitive development & then a second paragraph beginning with “Over the past years”. Perhaps the second paragraph should begin with some general description of the association between stunting and cognition, which will serve to alert the reader as to the purpose of the second paragraph (currently this is mentioned only at the end).
LINE 71- suggest revising to “Undernutrition is cause of poverty because of its impacts on cognitive development, and poverty is cause of undernutrition. Education is a powerful agent in disrupting this vicious cycle”.
RESPONSES TO COMMENTS #9, #10, #11, #12: As a reviewer, my comments reflect questions I believe other readers might share. Therefore, please incorporate your responses into the manuscript, rather than only responding only in your comments to me. You should provide a citation describing the cross-cultural validity of the HHS.
Author Response
Thank you for your constructive feedback. We have addressed all further points of criticism, and hope that we were able to modify the manuscript to your satisfaction.

This manuscript is a resubmission of an earlier submission. The following is a list of the peer review reports and author responses from that submission.
Round 1
Reviewer 1 Report
Stunting in childhood compromises the long-term cognitive and academic potentials of LMIC population. In this manuscript, the authors estimated the prevalence of stunting and evaluated risk factors associated with academic achievement and cognitive function among primary schoolers in marginalized South African communities. The identification of gender-specific risk factors are of substantial clinical and social importance, and offers valuable implications for future research and policymaking. Overall the study is well designed and data properly interpreted. I have only the following minor points that the authors can consider commenting on or discussing in more details.
- With the nutrition transition to a Western diet, low-to-middle-income countries are suffering from a mixed burden of undernutrition and overweight. It would be interesting if the authors could discuss how such forms of malnutrition differently impact the stunting prevalence and cognitive/academic performance in school children.
- Extensive government and NGO efforts have been devoted to fighting childhood malnutrition in South Africa. It would be interesting to compare student outcomes from local schools with different food environment. For example, do free/subsidized meal programs help relieve food insecurity, increase dietary diversity and improve the performance of children from low-to-middle-income families?
- Typo in L261: “incongruent stimuli stimuli”.
Author Response
Thank you for your encouraging and appreciative feedback. In the revised manuscript, we considered the mentioned minor points. Changes marked in the revised manuscript with yellow background color.
Sincerely,
Johanna Beckmann (on behalf of all authors)

Reviewer 2 Report
The work objectives are important and in line with the health priorities established by the Sustainable Development goals, especially in the context of Low- and Middle-income countries. In general, the manuscript is well written and present few typographical errors. However, I highlighted major and minor revisions that are needed to be addressed.
- Title
The title structure is not suitable and should indicate the study design, so I suggest: Assessment of academic achievement and cognitive function relationship with stunting and associated risk factors: A conducted cross-sectional study among South African primary school children
- Abstract
- Please add the study settings
- Line 22, please specify WHO references (2006)
- Line 25, linear regression instead of mixed linear regression
- Specify how the diet diversity was measured.
- Line 25, please indicate the 95% CI of the stunting prevalence
- Lines 25-27, indicates the regression coefficient
- Lines 42-44 should be rephrased
- Line 44 add a reference for the nutrition transition occurring in SA
- Line 46, long term consequences of early life adversity in form of malnutrition….
- Line 55, please standardize the reference (Pabalan et al., 2018)
- Reference 9 is good, but at this level a review (https://www.ncbi.nlm.nih.gov/pmc/articles/PMC6917415/) or a meta-analysis is better.
- Line 61, please insert "the vicious cycle of …"
- I think that the introduction is lacking data from previous studies about stunting and associated risk factors among the same age class as well as some information about the priorities of the country’s nutrition policy.
- Methods
- Line 121, why exactly three hours fasting period? give a reference.
- Please add details about the handling of blood samples and the apparatus/technic used for the hemoglobin quantification.
- Lines 129, 130 and 133 used standard form for references
- Line 134 add the kappa coefficient to mirror the concordance between the 2 technicians.
- Line 120-121 “Children were asked to have a three-hour fasting period before testing hemoglobin.” This sentence should be placed under the section 2.9
- Section 2.9. please indicated the coefficient of variation refering to the hemoglobin assay reproductibility.
- To my knowledge the “mixed linear regression” is a more common terminology used for longitudinal data analysis (containing fixed and random effects). I assume that the authors are referring to the classic linear regression. I suggest to remove "mixeed" across the manuscript.
- Did the authors adjust to the SES for the regression analysis? Please explain clear the regression model(s) used in this study by identifying the exposures and covariates. Also, you need to specify if adjusted analyses only are displayed or not?
- Results
- Section 3.1: Table 3, the data for SES was collected for 855 participants so that missing values reach 37.5%. As regard the high rate of missing data (>10%), I suggest imputing the data set (in the best case) or at least to acknowledge this in the section 3.1. At this high rate of missing, from my standing point I do recommended data imputation instead of full maximum likelihood estimator (probably, unless the auxiliary variables have a good correlation with the SES).
- Section 3.2: please report the confidence intervals for every prevalence.
- Table 1: please add the prevalence of anemia.
- Table 3: indicate that being 'not stunted' and 'no STH infection' are the reference categories
- Table 3: Are these results derived from a crude or adjusted analysis? Assuming that these are adjusted coefficients, I would like to draw the attention of authors that there is a collinearity issue here as the stunting is a categorization based on BMI-for-age z score.
Author Response
Thank you for your encouraging and appreciative feedback. We have tried to address all major and minor points of criticism, and hope that we were able to modify the manuscript to your satisfaction. Changes marked in the revised manuscript with yellow background color.
Sincerely,
Johanna Beckmann (on behalf of all authors)

Reviewer 3 Report
Overall Comments
This paper has may nice elements: I am appreciative of the description of quality-control for STH determination, the presence of a citation to back up the SES score determination, and other details that imply care taken during data collection. I also thought that the analyses were appropriate, although, as many of the factors considered might be inter-related, I also believe that more discussion of possible co-linearity, as well as the expected relationships between variables, could strengthen the work as well. The introduction, and to some extent the conclusion, might benefit from further editing & a further developed discussion of the existing literature.
Major Comments
More consideration might be paid to the pathways by which the risk factors considered in this analysis may interact. For example, how do poverty, food security, dietary diversity hemoglobin, and cognitive function relate to one another? If poverty leads food insecurity, which in term causes suboptimal function, it might be logical not to include all of these in the same model -at least not without careful consideration of the purported pathways to guide variable inclusion.
I also think that more should be said about the age of the children, and the extent to which the variables analyzed here reflect current insults or are thought to act as proxies for insults that may occurred in the past. Stunting, for instance, would seem to be a signal of early child deprivation, while hemoglobin is more likely a measure of current, rather than past status. Therefore, the lack of association here seems more likely to be related to the cross-sectional survey design than to a lack of biological association.
The reason that children of this age were selected should be provided.
The reason that these specific outcome measures were selected, among other possible tests or measurements, should also be provided. The underlying domains represented by the tests could be clarified, as could the theorized relationship between these domains. The age at which each of these domains develops might also be helpful to interpreting results: are these domains that develop at around this age, or are they domains that are expected to have developed earlier, during the time of stunting?
Minor Comments
LINE 23 – “Academic performance via school grades”
LINE 24 – “STH infection”, or “the prevalence of STH infection” not “STH infection rate” as rate implies incidence, which this cross-sectional survey cannot measure.
LINE 25 – “Mixed model” – please define the random intercept (school-class) in the abstract & methods section– it is currently only found in the discussion (LINE 381)
LINE 28 – “The prevalence of stunting”
LINE 30 – “Promoting nutrition may help…”
LINE 45 - I have some concerns about the author’s statement that a transition towards a “high-energy and fat diet” may be leading to child stunting. In general, with the nutritional transition, we do see reductions in the prevalence of stunting, although other forms of child undernutrition, such as anemia, may persist. Understanding trends in stunting over the past decade might be informative.
LINE 61 – Undernutrition is a cause of poverty because of its impacts on cognitive development
LINE 80 – Please describe the intervention that the overall study is testing.
LINE 105- If data was collected at the school yard during regular school hours, how is it that family SES was assessed by administering a questionnaire to caregivers – did they visit the school yard with their children at the time of the interview?
LINE 148 – The HHS is well-designed to measure severe food insecurity but may miss the mild-to-moderate food security that one might expect to be relatively more common in a community setting like this one. Please justify why this instrument was selected.
LINE 163- can you please clarify if risk factors were measured at the same tm as the end-of-year results, or before? If before, how long before?
LINE 387- Was any information about the child’s prior laptop usage noted?
Author Response
Thank you for your positive and constructive feedback. This is very much appreciated and helped to improve our manuscript. We have tried to address all mentioned comments, and hope that we were able to modify the manuscript to your satisfaction. Changes marked in the revised manuscript with yellow background color.
